# FGF9, a Potent Mitogen, Is a New Ligand for Integrin αvβ3, and the FGF9 Mutant Defective in Integrin Binding Acts as an Antagonist

**DOI:** 10.3390/cells13040307

**Published:** 2024-02-07

**Authors:** Chih-Chieh Chang, Yoko K. Takada, Chao-Wen Cheng, Yukina Maekawa, Seiji Mori, Yoshikazu Takada

**Affiliations:** 1Department of Dermatology, University of California, Davis School of Medicine, Sacramento, CA 95817, USA; imsidchang@gmail.com (C.-C.C.); yoktakada@ucdavis.edu (Y.K.T.); 2Department of Biochemistry and Molecular Medicine, University of California, Davis School of Medicine, Sacramento, CA 95817, USA; 3Graduate Institute of Clinical Medicine, College of Medicine, Taipei Medical University, Taipei 110, Taiwan; ccheng@tmu.edu.tw; 4Department of Medical Technology, Faculty of Health Science, Morinomiya University of Medical Sciences, Osaka 536-0025, Japan; 20176057@s.morinomiya-u.ac.jp (Y.M.); s-mori@morinomiya-u.ac.jp (S.M.); 5Department of Molecular Pathology, Division of Health Sciences, Graduate School of Medicine, Osaka University, Osaka 565-0871, Japan

**Keywords:** FGF9, integrin, dominant-negative effect, signaling, FGFR3, mutagenesis, docking simulation

## Abstract

FGF9 is a potent mitogen and survival factor, but FGF9 protein levels are generally low and restricted to a few adult organs. Aberrant expression of FGF9 usually results in cancer. However, the mechanism of FGF9 action has not been fully established. Previous studies showed that FGF1 and FGF2 directly bind to integrin αvβ3, and this interaction is critical for signaling functions (FGF–integrin crosstalk). FGF1 and FGF2 mutants defective in integrin binding were defective in signaling, whereas the mutants still bound to FGFR suppressed angiogenesis and tumor growth, indicating that they act as antagonists. We hypothesize that FGF9 requires direct integrin binding for signaling. Here, we show that docking simulation of the interaction between FGF9 and αvβ3 predicted that FGF9 binds to the classical ligand-binding site of αvβ3. We show that FGF9 bound to integrin αvβ3 and generated FGF9 mutants in the predicted integrin-binding interface. An FGF9 mutant (R108E) was defective in integrin binding, activating FRS2α and ERK1/2, inducing DNA synthesis, cancer cell migration, and invasion in vitro. R108E suppressed DNA synthesis and activation of FRS2α and ERK1/2 induced by WT FGF9 (dominant-negative effect). These findings indicate that FGF9 requires direct integrin binding for signaling and that R108E has potential as an antagonist to FGF9 signaling.

## 1. Introduction

Integrins are a superfamily of cell adhesion receptors that recognize extracellular matrix (ECM) ligands, cell surface ligands, and small soluble ligands (e.g., growth factors) [1]. It has been established that integrins are critically involved in growth factor signaling through integrin–growth factor crosstalk [2]. The first indication of the role of integrins in growth factor signaling is that antagonists of integrin αvβ3 suppressed FGF2-induced angiogenesis [3]. The specifics of this crosstalk are, however, unclear. Currently, popular models of integrin and growth factor crosstalk suggest that integrins promote growth factor signals through the interaction of integrins with the extracellular matrix [4].

The fibroblast growth factor (FGF) family is comprised of 22 members that can be divided into seven subfamilies in mammals [5]. FGF9 (FGF9 subfamily) is a secretory protein that was first isolated from human glioma cells [6]. Four isoforms of FGF receptors mediate FGF9’s biological effects [5]. FGF9 binding to its receptor causes receptor dimerization and activates different signal transduction cascades [7,8]. FGF9 is involved in a variety of complex responses in human or animal development. Mice lacking *Fgf9* have hypoplastic lungs [9], sex reversal [10], impaired germ cells [11], impaired skeletal growth [12], defective cardiomyocyte growth, and impaired inner ear development [13,14].

Although FGF9 messenger RNA (mRNA) is ubiquitously expressed in embryos, FGF9 protein expression is generally low and restricted to a few adult organs [15]. Aberrant activation of FGF9/FGFR signaling is associated with cancers [16,17,18,19]. FGF9 has oncogenic activity and is involved in the progression of cancers in the lung [15], stomach [20], colon, testis [16], and ovary [21]. FGF9 promotes epithelium and mesenchyme proliferation in the lungs [22]. Also, FGF9 enhances the cell proliferation and invasive ability of prostate cancer cells [23] and ovarian cancer [21]. Overexpression of FGF9 can promote tumor growth and liver metastasis of mouse Lewis lung carcinoma via EMT induction [24]. In addition, elevated FGF9 expression is associated with poor prognosis in non-small-cell lung cancer (NSCLC) [25].

Previous studies showed that FGF1 signaling requires direct integrin binding and the subsequent integrin–FGF1–FGFR1 ternary complex formation is required for signaling functions, and an FGF1 mutant defective in integrin binding (R50E) was defective in signaling functions [26] and suppressed FGF1 signaling induced by WT FGF1, acting as an antagonist of FGFR [27]. FGF1 mutants defective in integrin binding strongly blocked angiogenesis in vitro and tumor growth in vivo [27,28]. Similar results were obtained in FGF2 [28]. It is unclear if FGF9 requires crosstalk with integrins for signaling. We hypothesized that FGF9 directly binds to integrins and that FGF9 mutants defective in integrin binding act as antagonists for FGF9 signaling.

In the present study, we showed that FGF9 directly binds to integrin αvβ3. Docking simulation predicted that FGF9 would bind to the classical ligand-binding site of integrin αvβ3. We introduced point mutations in the predicted integrin-binding interface of FGF9. An FGF9 mutant R108E was defective in integrin binding and signaling functions. We found that R108E suppressed signaling induced by WT FGF9, indicating that R108E is a dominant-negative antagonist of FGF9. Thus, we propose that R108 has therapeutic potential and the integrin–FGF9 interaction is a novel therapeutic target.

## 2. Materials and Methods

### 2.1. Materials

DLD 1 human colorectal cancer and Colon26 mouse colon cancer cells were cultured in Dulbecco’s Modified Eagle Medium (DMEM) (Gibco, Bristol, RI, USA) containing 10% fetal bovine serum, 100 U/mL penicillin, and 100 μg/mL streptomycin. NIH3T3 mouse embryonic fibroblasts, which were obtained from Bioresource Collection and Research Center, Taiwan, were cultured in DMEM containing 10% bovine serum (Gibco, Bristol, RI, USA) and MycoZap (LONZA, Stein, Switzerland) in an atmosphere of 95% air, 5% CO_2_. The antibodies were purchased from the following sources: rabbit anti-FRS2α (ProteinTech, Rosemont, IL, USA), rabbit anti-phospho-FRS2α (Tyr-196) (Cell Signaling Technology, Danvers, MA, USA), rabbit anti-p44/42 MAPK (ERK1/2) (Cell Signaling Technology), anti-phospho-p44/42 MAPK (p-ERK1/2) (Thr-202/Tyr204) (Cell Signaling Technology), rabbit anti-FGFR1 (Cell Signaling Technology), rabbit anti-FGFR2 (Sigma Aldrich, St Louis, MO), and rabbit anti-FGFR3 (Novus Biologicals, Centennial, CO, USA). cRGDfV was obtained from Enzo Life Sciences Inc. (Lausen, Switzerland).

### 2.2. Synthesis of FGF9

The cDNA fragment encoding human FGF9 (LGEVGNYFGVQDAVPFGNVPVLPVDSPVLLSDHLGQSEAGGLPRGPAVTDLDHLKGILRRRQLYCRTGFHLEIFPNGTIQGTRKDHSRFGILEFISIAVGLVSIRGVDSGLYLGMNEKGELYGSEKLTQECVFREQFEENWYNTYSSNLYKHVDTGRRYYVALNKDGTPREGTRTKRHQKFTHFLPRPVDPDKVPELYKDILSQS) was amplified by PCR using full-length human FGF9 cDNA as a template and subcloned into the BamH1/EcoR1 site of a pET28aAmp vector, in which the kanamycin resistance gene was replaced with an ampicillin resistance gene. Site-directed mutagenesis was performed using the QuickChange method [29]. The existence of FGF9 mutations was checked by DNA sequencing. The WT FGF9, R108E, K129E, and K154E mutants were expressed in *E. coli* strain BL21(DE3) by isopropyl β-d- thiogalactoside (IPTG) induction and synthesized as insoluble proteins. The His-tags of proteins were used to purify proteins using Ni-NTA affinity chromatography in denaturing conditions (8 M urea). The Ni-NTA resin was washed with 0.5% Triton X-114 (Sigma Aldrich) to eliminate endotoxin before eluting the bound protein. The purified proteins were eluted in 250 mM imidazole/8 M urea. Purified proteins were diluted into refolding buffer (100 mM Tris-HCl, pH 8.0, 400 mM Arg, 2 mM EDTA, 0.5 mM oxidized glutathione, 5 mM reduced glutathione, and PMSF) on ice. The dilution was kept for 16 h at 4 °C with a slow stirring movement. Then, the proteins were concentrated by ultrafiltration. Around 4 to 5 milligrams of purified proteins were obtained from one liter of bacterial culture. Purified protein concentration was determined by measuring A280 and using the Bio-Rad protein assay.

### 2.3. Docking Simulation

A docking simulation of the interaction between FGF9 (PDB code 1IHK) and integrin αvβ3 (PDB code 1L5G) was performed using Autodock 3.05. In the present study, we used the headpiece (residues 1–438 of αv and residues 55–432 of β3) of αvβ3 (open-headpiece form, 1L5G.pdb). Cations were not present in αvβ3 during the docking simulation [26,30].

### 2.4. Binding of Soluble Integrin αvβ3 to FGF9

ELISA-type binding assays were performed as previously described [31]. Briefly, wells of 96-well microtiter plates were coated with FGF9 by incubating for 30 min at room temperature and the remaining protein-binding sites were blocked with BSA (heat-treated). Wells were incubated with soluble αvβ3 (1 μg/mL) in Tyrode-HEPES buffer in 1 mM Mn^2+^ and incubated for 1 h at room temperature. After washing the wells with the same buffer, bound αvβ3 was quantified using anti-β3 (mAb AV10), HRP-conjugated anti-mouse IgG, and peroxidase substrate.

### 2.5. FGF9 Binding to the FGFR1 D2D3 Fragment and Heparin

The ligand-binding site of FGFR1 (the immunoglobulin-like D2 and D3 domains, amino acid residues 140–365) was synthesized as described [26]. Briefly, pET21a encoding the cDNA fragment in the BamHI/XhoI sites of the vector was used to transform BL21 (DE3). The protein was expressed as an insoluble protein and refolded. The refolded protein was purified by affinity chromatography on heparin–Sepharose to enrich the properly folded protein. Bound protein was eluted with 1 M NaCl.

### 2.6. Binding of R108E to Heparin

We incubated purified WT and mutant FGF9 (100 μg each) in H_2_O with heparin–Sepharose (200 μL) in a small column and washed the beads with H_2_O (7 mL). Two hundred microliters of 1.5 M NaCl were added to the beads and eluted materials were collected (repeated once). The beads were heated in SDS sample buffer (200 μL) at 95C for 10 min and eluted materials were collected. The eluted materials were analyzed by SDS-PAGE and proteins were stained with Coomassie Brilliant Blue.

### 2.7. Cell Migration Assay

A polycarbonate filter of 8 µm pore size in the Chemotaxicell chamber (Kurabo, Osaka, Japan) was used to test cell migration. The lower side of the filter was coated with 10 µg/mL fibronectin (Asahi Glass, Tokyo, Japan) for 1 h at room temperature. After washing, the chamber was placed into a 24-well cell culture plate, and the lower portion of the plate was filled with serum-free DMEM containing 50 ng/mL WT FGF9 or R108E FGF9. Cells were plated on the upper side of the chamber and incubated at 37 °C for 6 h. Cells were fixed and visualized by crystal violet staining. The uncoated upper side of each filter was wiped with a cotton swab to remove cells that had not migrated through the filter. Migrated cells were counted from the digital images of the stained cells to determine the mean number of cells counted per field. Results were expressed as means ± SDs of the cell number.

### 2.8. Invasion Assay

Invasion assays were performed in a filter of 8 µm pore size in the Chemotaxicell chamber coated with 100 µg/mL growth factor-reduced Matrigel (BD Biosciences, San Jose, CA, USA) for 3 h at 37 °C and blocked with 0.1% bovine serum albumin in PBS. Cells were suspended in serum-free DMEM containing 0.1% BSA and plated in the upper chamber. The lower chamber was filled with DMEM containing 0.1% BSA and 50 ng/mL recombinant human WT FGF9 or R108E FGF9. Then, cells were allowed to migrate for 24 h. The top side of the filters was wiped with cotton swabs, fixed, and stained with 0.1% crystal violet. Images were taken by a digital camera and counted using the cell-counting function of Image J software (version 1.50).

### 2.9. BrdU Incorporation Assay

DNA synthesis or replication was monitored by measuring the Cell Proliferation ELISA BrdU kit (Roche, Indianapolis, IN, USA). Cells were cultured on a Nunc 96-well plate (Thermo Scientific, Waltham, MA, USA) at a density of 2 × 10^3^ cells/well. After 24 h, cells were rendered quiescent by incubation in serum-free medium for 16 to 18 h and then stimulated with either WT FGF9 or FGF9 mutations for 16 h. BrdU labeling solution was added to each well concomitantly and those cells were incubated for 2 h in a CO_2_ incubator at 37 °C in the presence of BrdU. Incorporated BrdU was detected using a monoclonal mouse anti-BrdU antibody conjugated with HRP and tetramethylbenzidine (TMB).

### 2.10. Statistical Analysis

Data processing was performed using Prism 7 software (GraphPad, Boston, MA, USA). All results were expressed as mean ± standard deviations. The statistical analysis of differences between two groups was performed by an unpaired Student’s *t* test and the difference between multiple groups was performed by a one-way analysis of variance (ANOVA). *p* < 0.05 was considered to indicate statistical significance.

## 3. Results

### 3.1. FGF9 Directly Binds to Integrin αvβ3

We studied the ability of FGF9 to bind to integrin αvβ3 in ELISA-type binding assays. The wells of a 96-well microtiter plate were coated with FGF9, and the remaining protein-binding sites were blocked with BSA. We found that the soluble integrin αvβ3 (extracellular domains) bound to immobilized FGF9 in a dose-dependent manner in 1 mM Mn^2+^ (Figure 1a), as predicted by docking simulation. Soluble integrin αvβ3 did not bind to wells that were coated only with BSA (which served as a negative control), and therefore FGF9 binding to αvβ3 is specific. Cyclic RGDfV, an inhibitor specific to αvβ3, suppressed the binding of αvβ3 to FGF9, indicating that αvβ3 specifically bound to FGF9 (Figure 1b). The binding of FGF9 to integrin αvβ3 required Mn^2+^, but Ca^2+^, Mg^2+^, or EDTA (all 1 mM) did not support FGF9 binding to integrin αvβ3 (Figure 1c). Also, heat treatment reduced FGF9 binding to αvβ3, indicating that FGF9 should be properly folded. These findings indicate that FGF9 is a ligand for integrin αvβ3.

To study how FGF9 binds to integrin αvβ3, we performed a docking simulation of the interaction between αvβ3 (PDB code 1L5G) and FGF9 (PDB code 1IHK) using AutoDock 3. We performed 50 independent dockings, and the obtained poses were clustered (RMSD < 2.0). The pose in the first cluster (docking energy −23 Kcal/mol, the first cluster) was selected for further analysis. The simulation predicted that FGF9 binds to the classical ligand-binding site of αvβ3 at high affinity (Figure 2a). When the 3D structure of the FGF9-FGFR1 complex (5W59.pdb) was superposed with the αvβ3-FGF9 docking model, there was little or no steric hindrance (Figure 2b). This predicts that the FGFR1-binding site and the integrin-binding site are distinct and that the FGF9-binding site overlaps with that of FGF1 [26], indicating that FGF9 binds to the classical ligand-binding site (site 1) of integrins. Amino acid residues that are predicted to be involved in FGF9–integrin αvβ3 interaction are shown in Table 1.

### 3.2. Generation of Integrin-Binding Defective FGF9 Mutants

We chose Arg108, Lys129, and Lys154 in the predicted integrin-binding interface of FGF9 for mutagenesis studies (Figure 3a, Table 1). The Arg108-to-Glu (the R108E mutation) and the K154E mutations, and the K129E mutation to a lesser extent, reduced the binding of soluble αvβ3 to FGF9 in ELISA-type binding assays (Figure 3b). This result indicated that the R108E and K154E mutations effectively suppressed the binding of FGF9 to αvβ3. FGF9 has been shown to induce strong dose-dependent mitogenic activity in NIH3T3 cells [32]. This justifies studying whether FGF9 mutations affect DNA synthesis in NIH3T3 cells (Figure 3c). R108E completely lost the ability to induce BrdU incorporation, but K129E and K154E mutants were partly defective, indicating that direct binding to integrin αvβ3 is critical for the signaling functions of FGF9. We selected R108E for further analysis.

We studied if the R108E mutation affects FGFR1 binding using the FGFR1 D2D3 fragment in ELISA-type binding assays. FGF9 WT and FGF9 R108E bound to immobilized FGFRD2D3 at a comparable level (Figure 4a), indicating that the R108E mutation did not detectably affect FGFR1 binding.

FGF9 is known to bind to heparin but it is possible that the R108E mutation affects heparin binding. To study this possibility, we incubated partially purified WT and mutant FGF9 (100 μg each) in H_2_O with heparin–Sepharose (200 μL) in a small column and washed the beads with H_2_O (7 mL). Two hundred microliters of 1.5 M NaCl were added to the beads and eluted materials were collected (repeated once). The beads were then heated in SDS sample buffer (200 μL) at 95C for 10 min and eluted materials were collected. We found that FGF9 WT and R108E bound to the heparin beads and were not eluted by 1.5 M NaCl (Figure 4b). These findings indicate that FGF9 strongly binds to heparin and the R108E mutation did not detectably affect the heparin-binding ability of FGF9.

### 3.3. R108E Is Defective in Inducing Migration and Invasion of Colon Cancer Cells

FGF9 is a potent chemoattractant. We found that R108E was defective in inducing cell migration (Figure 4c,d) and invasion of DLD1 and Colon26 cells (Figure 4e,f).

### 3.4. R108E Suppresses Activation of FRS2 and ERK1/2 and DNA Synthesis Induced by WT FGF9 (Dominant-Negative Effect) in NIH3T3 Cells

If integrin binding to FGF9 and integrin–FGF9–FGFR ternary complex formation is required for FGF9 signaling, it is expected that the FGF9 R108E mutant defective in integrin binding is expected to be dominant-negative, as in the case of FGF1 and FGF2 [27,28]. We studied whether R108E suppresses FRS2α and ERK1/2 phosphorylation induced by WT FGF9 in NIH3T3 cells. NIH3T3 cells were incubated with WT FGF9 and/or R108E for 30 min. We found that 20-fold excess R108E blocked FRS2α and ERK1/2 activation induced by WT FGF9 (Figure 5a–c). We examined whether R108E suppresses DNA synthesis induced by WT FGF9 in NIH3T3 cells in BrdU incorporation assays. R108E did not induce DNA synthesis. We found that excess (20x) R108E suppressed DNA synthesis induced by WT FGF9 (Figure 5d). These findings suggest that R108E is a dominant-negative mutant of FGF9.

## 4. Discussion

### R108E (FGF9 Antagonist) Has Potential as Therapeutic in Cancer

FGF9 has been implicated in the pathogenesis of cancer through its high-affinity receptor FGFR3c. FGF9 is over-expressed in cancer-associated fibroblasts (CAFs) and mediates communication with cancer cells [33]. This indicates that FGF9 is a major therapeutic target in cancer, and antagonists to FGF9 must be developed. In our previous studies, docking simulation predicted that FGF1 and FGF2 directly bind to integrins, and we developed dominant-negative FGF1 and FGF2 mutants, which have therapeutic potential [26,28]. The present study showed for the first time that FGF9 requires direct integrin binding for signaling functions. Docking simulation predicted that FGF9 binds to the classical ligand-binding site of αvβ3. We generated an FGF9 mutant defective in integrin binding (the R108E mutant). The position of the R108E mutation cannot be deduced from the comparison of the primary structures of FGF9 and FGF1/FGF2, probably because FGF1, 2, and 9 interact with integrins differently.

The R108E mutant was defective in signaling functions, including DNA synthesis, activation of ERK1/2 and FRS2α, cell migration, and invasion. This indicates that binding of FGF9 to integrin is required for FGF9 signaling. Notably, R108E is a dominant-negative mutant and effectively suppressed DNA synthesis, FRS2 phosphorylation, and ERK1/2 activation induced by WT FGF9. This suggests that R108E has potential as a therapeutic in diseases in which FGF9 is involved. Previous studies showed that the dominant-negative FGF1 mutant R50E bound to FGFR1 with an affinity comparable to that of WT FGF1 [26]. We predict that FGF9 induces integrin–FGF9–FGFR ternary complex formation on the cell surface, which is critical for FGF9 signaling. FGF9 binds to FGFR (high-affinity receptor) and integrins (low-affinity receptor). The loss of integrin binding is assumed to result in the loss of signaling functions but FGFR binding is retained, which makes it a potent dominant-negative antagonist of FGF9 signaling. Since FGF9 has oncogenic activity and is overexpressed in many cancers (see Introduction), FGF9 R108E should be useful as a potential therapeutic. It is highly likely that binding of FGF9 to FGFR is not sufficient for FGF9 signaling. It would be interesting to study if the R108E mutant of FGF9 suppresses cancer proliferation by inhibiting FGF9 signaling (e.g., through FGFR3) in future studies.

It is possible that FGF9 binds to integrins other than αvβ3. If this is the case, blocking αvβ3 using antagonists of αvβ3 may not be effective in blocking FGF9 signaling, since other integrins may replace the position of αvβ3. However, the FGF9 R108E mutant can block FGF9–integrin interaction regardless of integrin species.

Previous studies showed that several growth factors (e.g., FGF1, IGF1, fractalkine, and CD40L) require direct integrin binding and subsequent integrin–growth factor–cognate receptor ternary complex formation for signaling functions (ternary complex model), and growth factor mutants defective in integrin binding were defective in signaling functions, but they still bound to cognate receptors and acted as antagonists of growth factor signaling (growth factor decoys) [26,28,31,34,35]. We propose that integrins are common co-receptors of growth factors and the ternary complex model can be applied to many types of growth factor signaling. If this is the case, growth factor antagonists can be designed by screening growth factor mutants defective in integrin binding.

Basic fibroblast growth factor (bFGF, FGF2) and integrin α6β1 are important for maintaining the pluripotency of human pluripotent stem cells (hPSCs). We previously showed that FGF2 binds to integrins and the FGF2 K125E mutant is defective in integrin binding and in inducing signals [28]. It has recently been reported that integrin α6β1-FGF2-FGFR ternary complex formation is critical for maintaining the pluripotency of hPSCs [36]. The FGF2 K125E was incapable of inducing the hPSC properties, such as proliferation, ERK activity, and large focal adhesions at the edges of human induced pluripotent stem cells (hiPSCs) colonies. These findings suggest that the integrin-FGF interaction and resulting ternary complex formation is a potential novel target for drug discovery. Dominant-negative FGF9 R108E mutant would be a potential therapeutic and useful to study the role of FGF9 in cancer.

## Figures and Tables

**Figure 1 cells-13-00307-f001:**
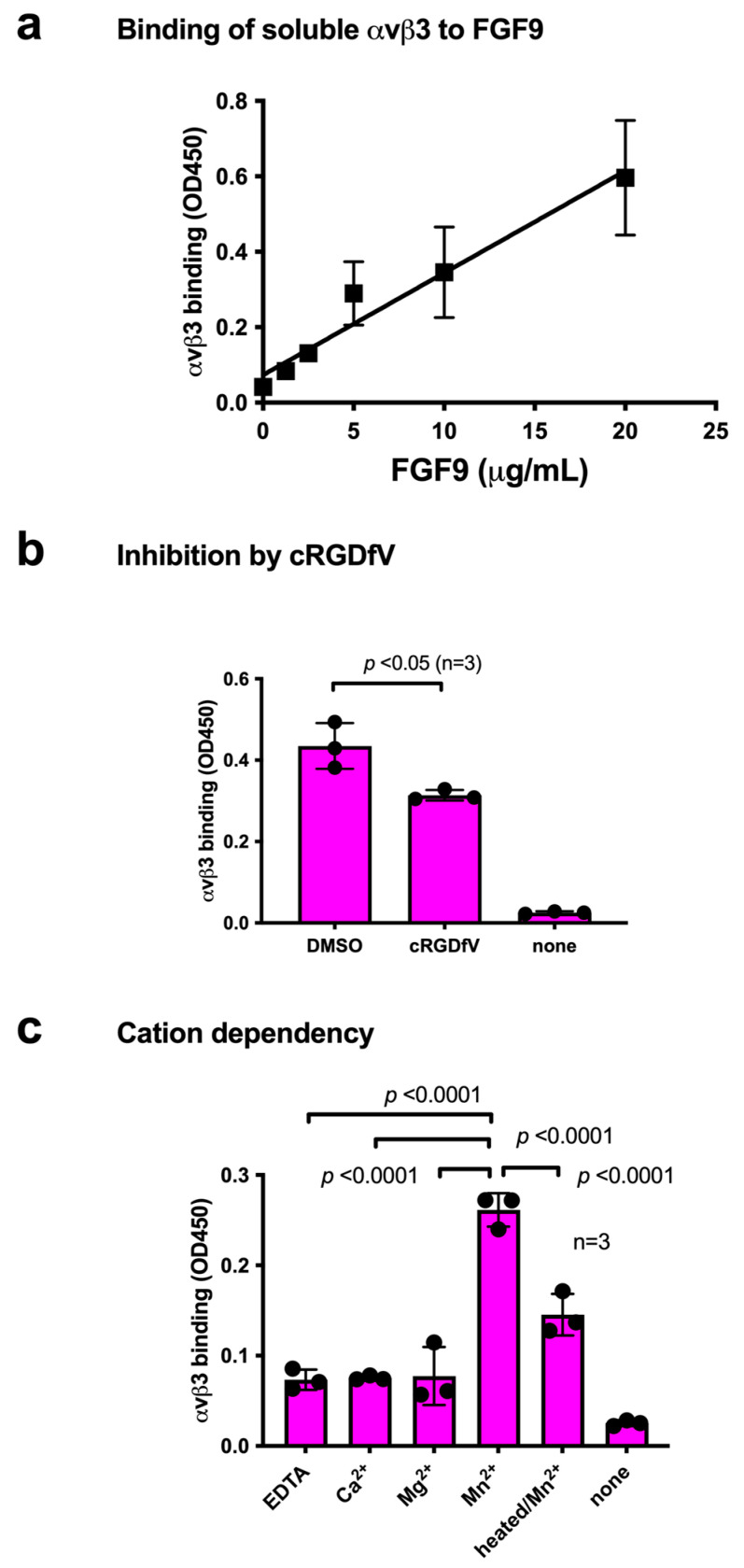
**Binding of soluble integrin αvβ3 to FGF9.** (**a**) Binding of FGF9 mutant to αvβ3 in ELISA binding assay. Wells of 96-well microtiter plate were coated with FGF9 and remaining protein-binding sites were blocked with BSA. Soluble αvβ3 (1 μg/mL) was added to wells and incubated for 1 h in Tyrode-HEPES buffer containing 1 mM Mn^2+^. Bound αvβ3 was quantified using anti-β3 and HRP-conjugated anti-mouse IgG. (**b**) Binding of soluble αvβ3 in the presence of cyclic RGDfV, a specific inhibitor for αvβ3. Data are expressed as means ± SDs of triplicate experiments. (**c**) The effect of cations and heat treatment on the binding of soluble integrin αvβ3 to FGF9. Cations (1 mM) were included in Tyrode-HEPES buffer. Data are expressed as means ± SDs of triplicate experiments.

**Figure 2 cells-13-00307-f002:**
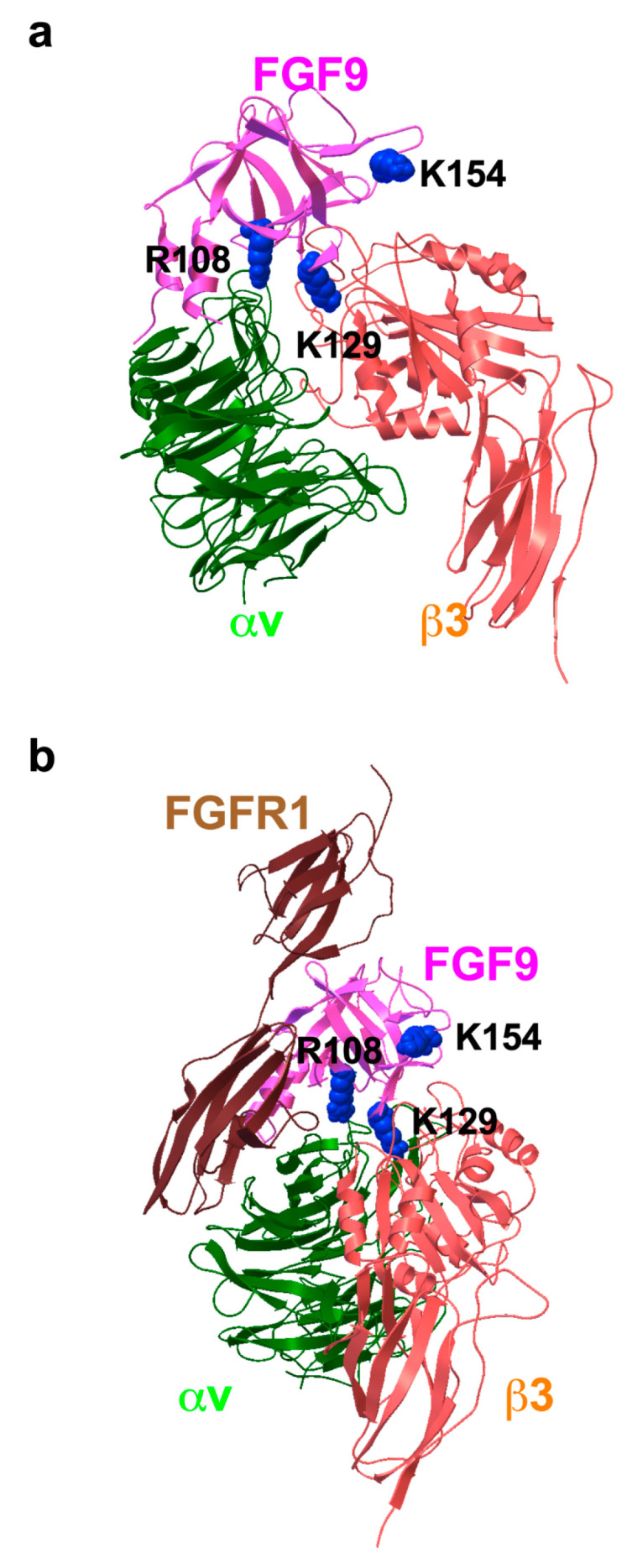
**Docking simulation of FGF9–integrin interaction.** Docking simulation of interaction between FGF9 and integrin αvβ3 predicted that FGF9 binds to αvβ3 at high affinity (docking energy −23 Kcal/mol). (**a**) The FGF9-αvβ3 docking model. (**b**) FGF9-FGFR1 complex was superposed to the FGF9-αvβ3 docking model. The simulation predicted that FGF9 binds to the classical ligand-binding site of αvβ3.

**Figure 3 cells-13-00307-f003:**
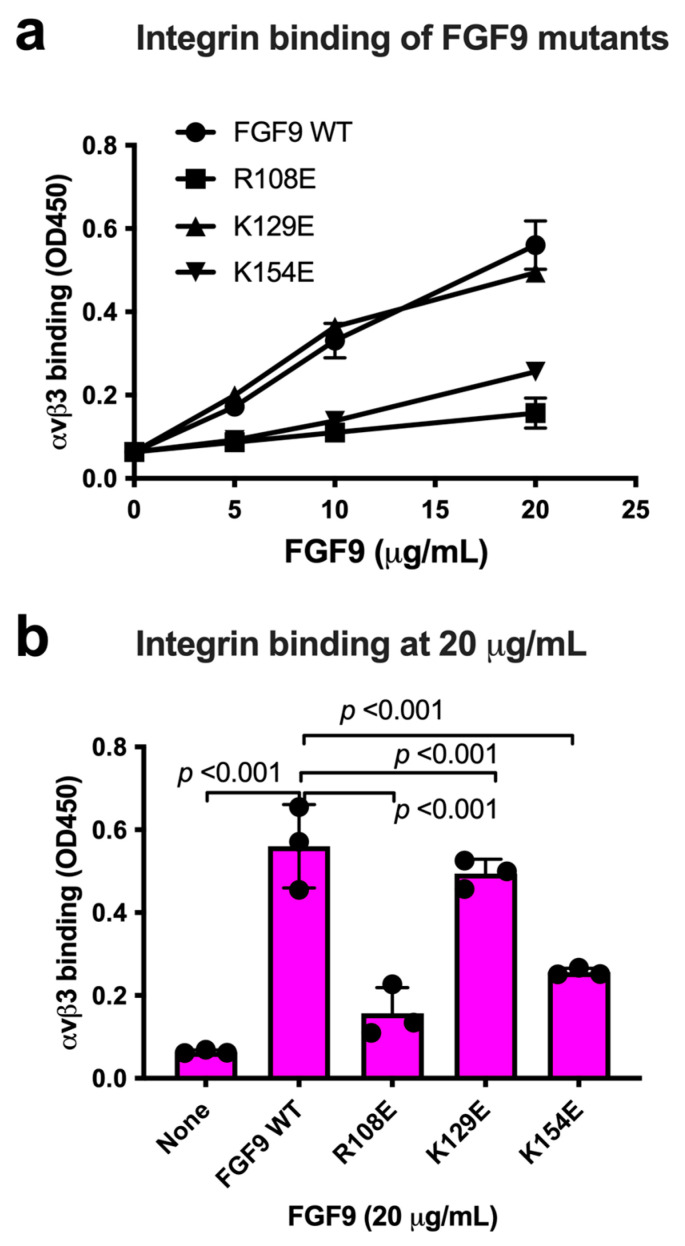
**Identification of FGF9 mutants defective in integrin binding.** (**a**) FGF9 mutants are defective in integrin binding. We chose several amino acid residues of FGF9 for mutagenesis studies. Arg108, Lys129, and Lys154 of FGF9 in the predicted integrin-binding site (Figure 2) were mutated to Glutamate. The ability of FGF9 mutants (the R108E, K129E, and K154E mutants) to bind to integrin was measured in binding assays using soluble αvβ3. Data are shown as means ± S.D. of triplicate experiments. (**b**) The binding of FGF9 mutants at 20 μg/mL is shown. Data are shown as means ± SDs of triplicate experiments. (**c**) Effect of FGF9 mutations on DNA synthesis. NIH3T3 cells were starved for 24 h and stimulated with FGF9 (WT and mutants at 100 ng/mL), and BrdU was added to the medium for the last 2 h of the incubation.

**Figure 4 cells-13-00307-f004:**
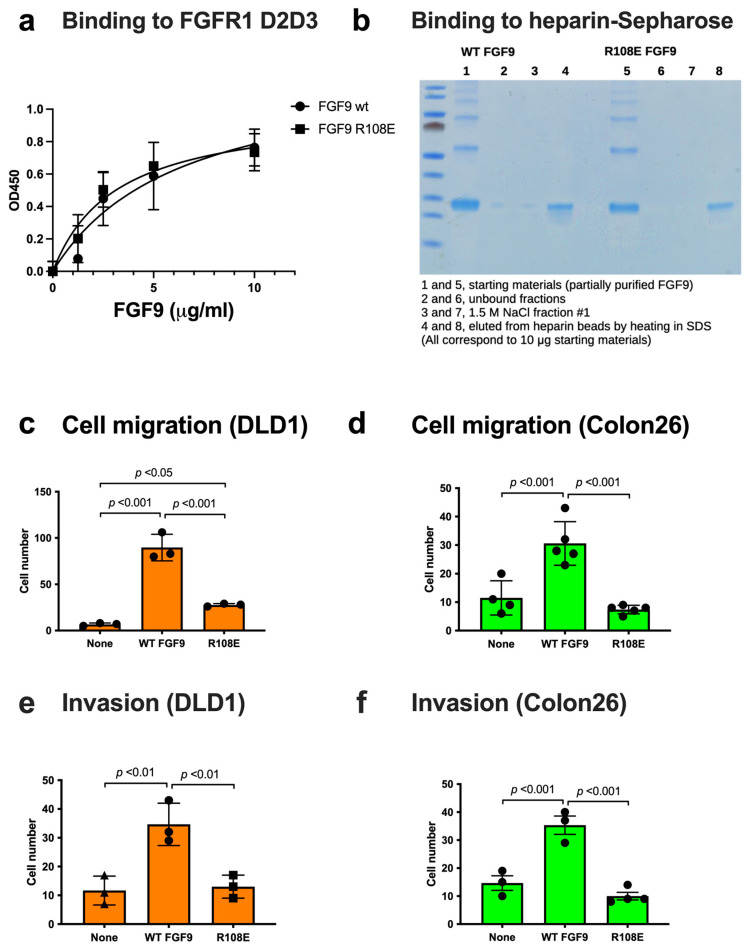
**FGF9 mutant defective in integrin binding (R108E) is defective in inducing cell migration and invasion.** (**a**) Binding of R108E to FGFR1. Wells of 96-well microtiter plate were coated with the FGFR1 D2D3 fragment and incubated with FGF9 and R108E. After washing with the binding buffer, bound FGF9 was quantified using HRP-conjugated anti-6His antibody. Data are shown as means ± SDs of triplicate experiments. (**b**) Binding of R108E to heparin; we incubated partially purified WT and mutant FGF9 (100 μg each) with heparin–Sepharose (200 μL) and washed with H_2_O (7 mL). The beads were incubated with 1.5 M NaCl (200 μL) twice and eluted fractions were collected. The beads were heated at 95C for 10 min in SDS sample buffer. The eluted materials were analyzed by SDS-PAGE and proteins were stained with Coomassie Brilliant Blue (10% of each fraction corresponding to 10 μg of the starting material was applied per lane). (**c**,**d**) Migration of DLD1 and Colon26 colon cancer cells to FGF9 was measured using Chemotaxicell chamber as described in the Methods section. Data are shown as means ± SDs of triplicate experiments. (**e**,**f**) Invasion of DLD1 and Colon26 colon cancer cells induced by FGF9 was measured using Chemotaxicell chamber coated with growth factor-depleted Matrigel as described in the Methods section. Data are shown as means ± SDs of triplicate experiments.

**Figure 5 cells-13-00307-f005:**
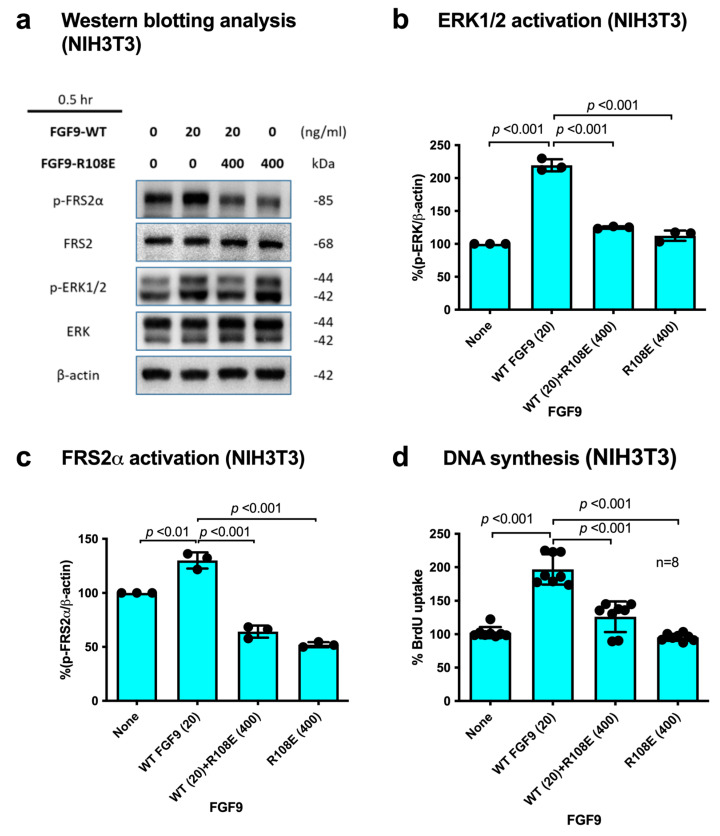
**R108E suppresses FRS2 and ERK1/2 activated by WT FGF9 in NIH3T3 cells.** (**a**) NIH3T3 cells were starved for 24 h and stimulated with WT FGF9 (20 ng/mL) and/or R108E (400 ng/mL). Cell lysates were analyzed by Western blotting using anti-phospho-FRS2α, anti-FRS2α, anti-phospho-ERK1/2, and anti-total ERK1/2. β-actin was used as an internal control. Density of the bands was quantified using ImageJ software and phospho-ERK1/2/β-actin (**b**) or phospho-FRS2α/β-actin (**c**) was calculated. Data are shown as means ± SDs of triplicate experiments. (**d**) NIH3T3 cells were starved for 24 h and stimulated with WT FGF9 (20 ng/mL) and/or R108E (400 ng/mL), and BrdU was added to the medium for the last 2 h of the incubation. Data are shown as means ± SDs of triplicate experiments.

**Table 1 cells-13-00307-t001:** Amino acid residues that are predicted to be involved in integrin αvβ3 and FGF9 interaction (site 1).

FGF9	β3	αv
Thr52, Asp53, Asp55, His56, Leu57, Pro78, Asn79, Gly80, Thr81, **Arg108**, Val110, Asp111, Ser112, Gly113, Leu114, Tyr115, Asn119, Glu120, Tyr125, Gly126, Ser127, Glu128, **Lys129**, Leu130, Thr131, Gln132, Glu133, Cys134, Asn151, Leu152, **Lys154**, Asp203, Leu204, Leu205, Ser206, Glu207, Ser208	Ser121, Tyr122, Ser123, Met124, Lys125, Asp126, Asp127, Leu128, Trp129, Tyr166, Cys177, Tyr178, Asp179. Met180, Lys181, Yhr182, Arg214, Asn215, Arg216, Asp217, Ala218, Pro219, Glu220, Asp251, Thr311, Glu312, Asn313, Leu333, Ser334, Met335, Asp336, Ser337	Asp146, Asp148, Ala149, Asp150, Tyr178, Trp179, Gln182, Asn205, Asn206, Gln207, Leu208, Ala209, Arg211, Thr212, Ala213, Gln214, Ala215, Phe217, Asp218, Asn260,

Amino acid residues within 0.6 nm between HBD and αvβ3 were selected using Pdb Viewer (version 4.1). Amino acid residues in HBD that were selected for mutagenesis are shown in bold.

## Data Availability

Data is available upon request.

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
