# Peer review of "FGF9, a Potent Mitogen, Is a New Ligand for Integrin αvβ3, and the FGF9 Mutant Defective in Integrin Binding Acts as an Antagonist"

_cells, 2024, doi:10.3390/cells13040307_

Round 1
Reviewer 1 Report
Comments and Suggestions for Authors
In this manuscript, Chang et al. tackle the question whether FGF9 can bind to integrin avb3, which has been previously identified to directly interact with FGF1 and FGF2. Using an ELISA-type binding assay, Chang et al. demonstrated that immobilized FGF9 binds to integrin avb3, which could be suppressed by the addition of cyclic RGDfV, an integrin avb3 inhibitor. The authors then attempt to identify several integrin-binding defective FGF9 mutants by utilizing site-directed mutagenesis, whereby the relevant amino acids were selected via docking simulations between FGF9 and integrin avb3. Two successful integrin-binding defective FGF9 mutants, FGF9 R108E and FGF9 K154E reduced the binding to integrin avb3 in an immobilized assay. Furthermore, the mutants showed suppressed activation of FRS2a and ERK1/2 through inhibition of phosphorylation as well as suppressed DNA synthesis relative to the FGF9 wildtype, indicating a dominant-negative effect of the FGF9 mutants. The introduction does an adequate job of scrutinizing the demand for this investigation and the current standing of the area. The authors outline the scientific approach starting from the simulation of FGF9-integrin avb3 docking sites to functional validating their hypothesis and elucidating that the FGF9 mutant reverts the cellular phenotype of FGF9 WT.
Major comments:
1. The authors do not provide a justification for their choice of assays to validate their findings. This leaves the reader with unanswered questions about why specific approaches were used and whether other functional assays could have been employed to strengthen their claims.
2. The study does not provide a comprehensive illustration of the dominant-negative mutant's intracellular signaling details. The paragraphs at lines 231 until 263 is too short and fails to provide sufficient information. It would be great if the authors could include another relevant cancer cell line and repeat the migration/invasion assay (see below).
3. The authors' focus on ERK1/2 and FRSa phosphorylation in NIH3T3 cells is not justified. as this aspect is particularly interesting due to the significant role of FGFs in signaling. The experiments should be repeated in the colon cancer cells with data in NIH3T3 provided as Supp Information.
4. Figure 1a: What is the negative control? This should be included.
5. Figure 2: the analysis requires more details
6. Figure 3c: the experiment should be repeated in colon cancer cells to make the findings stronger
7. Figure 4a: a loading control is missing
8. Figure 4b: total ERK bands do not seem to correspond to the p-ERK1/2 bands. If total ERK has not been detected after stripping the authors must show two different loading controls, one for each antibody and perhaps quantify the results
9. Figure 5a. do p-ERK1/2 and ERK1/2 come from the same blot? It does not seem the case.
10. Figure 5c-d: the quantification should be done over total ERK from the same blot and then over actin or over actin and then over total ERK1/2 over actin if p-ERK1/2 and ERK1/2 come from two different blots
11. Why do the authors use different concentrations of the WT and MUT FGF9 in their experiments?
Minor comments:
1. Line 84: what is the subtitle here?
2. The authors use clear language, but there are some typographic errors (line 189, 190, 230, 279).
3. The discussion references six previous works and attempts to relate the research findings to the current state of the field. The hypothesis of high- and low-affinity receptors is particularly compelling. The authors deduce that FGF9 strongly interacts with FGFR and only weakly binds to integrin avb3. This hypothesis would require more elucidation.
4. The authors suggest inhibiting FGF9 signaling and elucidating its role in cancer proliferation as a potential area for future studies, which would be fundamental to further understand the biological use of this dominant-negative FGF9 mutant. Discussing this point further would strengthen the manuscript.
Comments on the Quality of English Language
Some edits and changes are suggested
Author Response
In this manuscript, Chang et al. tackle the question whether FGF9 can bind to integrin avb3, which has been previously identified to directly interact with FGF1 and FGF2. Using an ELISA-type binding assay, Chang et al. demonstrated that immobilized FGF9 binds to integrin avb3, which could be suppressed by the addition of cyclic RGDfV, an integrin avb3 inhibitor. The authors then attempt to identify several integrin-binding defective FGF9 mutants by utilizing site-directed mutagenesis, whereby the relevant amino acids were selected via docking simulations between FGF9 and integrin avb3. Two successful integrin-binding defective FGF9 mutants, FGF9 R108E and FGF9 K154E reduced the binding to integrin avb3 in an immobilized assay. Furthermore, the mutants showed suppressed activation of FRS2a and ERK1/2 through inhibition of phosphorylation as well as suppressed DNA synthesis relative to the FGF9 wildtype, indicating a dominant-negative effect of the FGF9 mutants. The introduction does an adequate job of scrutinizing the demand for this investigation and the current standing of the area. The authors outline the scientific approach starting from the simulation of FGF9-integrin avb3 docking sites to functional validating their hypothesis and elucidating that the FGF9 mutant reverts the cellular phenotype of FGF9 WT.
Major comments:
- The authors do not provide a justification for their choice of assays to validate their findings. This leaves the reader with unanswered questions about why specific approaches were used and whether other functional assays could have been employed to strengthen their claims.
Response: We revised the text to provide more justifications.
- The study does not provide a comprehensive illustration of the dominant-negative mutant's intracellular signaling details.
Response: Probably the reviewer is requesting the mechanism in how integrin is involved in FGF9 signaling. We focus on the potential mechanism in which integrin-FGF9-FGFR ternary complex is required for FGF9 signaling. The present manuscript describes the dominant-negative FGF9, which will be useful for studying the role of integrins in FGF9 signaling inside the cells. We will need to study if integrin outside-in signaling is involved in future experiments.
The paragraphs at lines 231 until 263 is too short and fails to provide sufficient information.
Response: We included more information (lines 231-263).
It would be great if the authors could include another relevant cancer cell line and repeat the migration/invasion assay (see below).
Response: We started experiments in other cancer cell lines but realized that we do not have enough time in this revision schedule. We will study in the future experiments.
- The authors' focus on ERK1/2 and FRSa phosphorylation in NIH3T3 cells is not justified. as this aspect is particularly interesting due to the significant role of FGFs in signaling. The experiments should be repeated in the colon cancer cells with data in NIH3T3 provided as Supp Information.
Response: Recent paper showed that FGF9 shows potent mitogenic activity in NIH3T3 cell (reference 35). This justifies using this cell line for testing the mitogenic activity of FGF9 mutants.
- Figure 1a: What is the negative control? This should be included.
Response: Wells of 96-well micro titer plate were blocked with BSA after coating with FGF9. BSA only serves as a negative control.
- Figure 2: the analysis requires more details.
Response: we added explanations as much as possible.
- Figure 3c: the experiment should be repeated in colon cancer cells to make the findings stronger
Response: We were not able to accomplish this due to time limitation.
- Figure 4a: a loading control is missing.
Response: Fig. 4a was deleted since we do not have enough time to address comments. Also, Fig. 4a may not have new information and deletion of Fig. 4a may not affect the interpretation of our results.
- Figure 4b: total ERK bands do not seem to correspond to the p-ERK1/2 bands. If total ERK has not been detected after stripping the authors must show two different loading controls, one for each antibody and perhaps quantify the results.
Response: Fig. 4b was deleted since we do not have enough time to address comments. Also, Fig. 4b may not have new information and deletion of Fig. 4b may not affect the interpretation of our results.
- Figure 5a. do p-ERK1/2 and ERK1/2 come from the same blot? It does not seem the case.
Response: The total ERK blots were not stripped from the p-ERK blots. But all samples were split equally for p-ERK and total ERK and electrophoresed separately. We have replaced the total ERK and b-Actin blots with different exposure times to make it clear that the amount of loading is the same.
We have checked the original blot images (submitted as supplemental Fig S1). It is possible that we used a different set of images of ERK and phosphoERK for the Fig 5a. It would be possible to replace the images but it may not be necessary, since we submitted Fig. S1.
- Figure 5c-d: the quantification should be done over total ERK from the same blot and then over actin or over actin and then over total ERK1/2 over actin if p-ERK1/2 and ERK1/2 come from two different blots.
Response: The total ERK blots were not stripped from the p-ERK blots. But all samples were split equally for p-ERK and total ERK and electrophoresed separately. We have replaced the total ERK and b-Actin blots with different exposure times to make it clear that the amount of loading is the same.
- Why do the authors use different concentrations of the WT and MUT FGF9 in their experiments?
Response: We use vast excess (20x) FGF9 mutant to suppress WT FGF9 signaling. This would be probably a standard procedure.
Minor comments:
- Line 84: what is the subtitle here?
Response. Materials
- The authors use clear language, but there are some typographic errors (line 189, 190, 230, 279). Response: Errors were corrected.
- The discussion references six previous works and attempts to relate the research findings to the current state of the field. The hypothesis of high- and low-affinity receptors is particularly compelling. The authors deduce that FGF9 strongly interacts with FGFR and only weakly binds to integrin avb3. This hypothesis would require more elucidation.
- The authors suggest inhibiting FGF9 signaling and elucidating its role in cancer proliferation as a potential area for future studies, which would be fundamental to further understand the biological use of this dominant-negative FGF9 mutant. Discussing this point further would strengthen the manuscript.
Response: This is a nice suggestion. We included the discussion.
Reviewer 2 Report
Comments and Suggestions for Authors
In this manuscript a FGF9 variant with impaired binding to integrin αvβ3 was developed and characterized. The authors postulate that the interaction of these proteins is crucial to achieve signal transduction, which influences cell fate. Since FGF-integrin interactions are important for carcinogenesis processes, the development of new therapeutic strategies is highly desirable.
This is a well-experimental designed research article that follows through the hypothesis of the work. Although I found this work very interesting and valuable, I am asking the authors of the manuscript to respond to the following objections.
Main point:
The authors predict that there will be no steric conflict between integrin αvβ3 and FGFR1, but this has not been proven. Showing the lack of effect of a point mutation on the binding of FGF9 to the canonical receptor would confirm this assumption and significantly increase the value of the presented manuscript. Moreover, position R108 is located near the HBS of FGF9, and the introduced mutation significantly changes the charge of the side chain. Is it possible that the binding to heparans, which are very important for stabilizing the binding of FGF to FGFR, has been disturbed? Disruption of any of these interactions would translate into weaker binding of the ligand, which could further result in a lack of cellular response.
I am also curious whether the authors performed the same cellular experiments (signaling, BrdU, migration/invasion) in the presence of high heparin concentration, which would also confirm that the observed lack of biological activity of the R108E mutein is integrin-dependent.
Minor points:
The part of the sentence "R108E suppressed DNA synthesis induced by WT FGF9 and suppressed DNA synthesis and activation of FRS2α and ERK1/2 induced by WT FGF9" appears to be a repetition (line 35-37).
No information about cyclic RGDfV in the Materials and methods section.
It seems to me that in the results (3.2, line 213) in the description of the mutants there should be a reference to Fig. 3a instead of Fig. 1a.
The sentences " We found that R108E was defective in inducing cell migration (Figs. 4c and 4d). R108E was also defective in inducing invasion of DLD1 and Colon26 cells (Fig. 4e and 4f). These findings suggest R108E is defective in inducing migration and invasion in colon cancer cells" require rewording because it is a repetition (line 236-239).
Figure 4b requires densitometric analysis.
How do the authors explain the lack of FGFR1 detection in the DLD1 cell line, which has been repeatedly known for the expression of this receptor? Why is only one band visible for FGFR2 and FGFR3, while these receptors usually present 2-3 forms? Can authors add a mass marker? (Fig. 4a, 4b).
In Fig. 5, how do the authors explain the different molecular weights of pFRS2 and FRS2? The shift may be due to phosphorylation, but then two bands should be visible in FRS2 detection, whereas only the one corresponding to the mass of the unphosphorylated protein is visible.
In my opinion, the intensity of the pErk band (Fig. 5a) does not correspond to the presented densitometric data (for the R108E mutein, last line on WB).
Author Response
Comments and Suggestions for Authors
In this manuscript a FGF9 variant with impaired binding to integrin αvβ3 was developed and characterized. The authors postulate that the interaction of these proteins is crucial to achieve signal transduction, which influences cell fate. Since FGF-integrin interactions are important for carcinogenesis processes, the development of new therapeutic strategies is highly desirable.
This is a well-experimental designed research article that follows through the hypothesis of the work. Although I found this work very interesting and valuable, I am asking the authors of the manuscript to respond to the following objections.
Main point:
The authors predict that there will be no steric conflict between integrin αvβ3 and FGFR1, but this has not been proven. Showing the lack of effect of a point mutation on the binding of FGF9 to the canonical receptor would confirm this assumption and significantly increase the value of the presented manuscript.
Response: Docking simulation has limitation and docking data should be interpreted carefully. We studied if WT FGF9 and the R108E mutant bind to FGFR fragment (D2D3). They bound to FGFR1D2D3 to a comparable level. This suggests that FGFR1 binding is not affected by the R108E mutation (new Fig. 4a).
Moreover, position R108 is located near the HBS of FGF9, and the introduced mutation significantly changes the charge of the side chain. Is it possible that the binding to heparans, which are very important for stabilizing the binding of FGF to FGFR, has been disturbed? Disruption of any of these interactions would translate into weaker binding of the ligand, which could further result in a lack of cellular response.
Response: We studied if FGF9 R108E binds to heparin-Sepharose and found that WT FGF9 and R108E bound to heparin-Sepharose (new Fig. 4b). These findings suggest that heparin binding may not be affected by this mutation.
We started an additional experiment, in which proliferation of several cancer cells are affected by FGF9 R108E mutant. However, we realized that we do not enough time to finish this experiment considering the time limit of revision. We will try this later in future experiments.
I am also curious whether the authors performed the same cellular experiments (signaling, BrdU, migration/invasion) in the presence of high heparin concentration, which would also confirm that the observed lack of biological activity of the R108E mutein is integrin-dependent.
Response: We previously developed two FGF2 mutants defective in integrin binding. Both showed dominant-negative effect (e.g., in signaling, cell proliferation and angiogenesis). However, one of them was defective in heparin binding, but another one was not defective. So, we assume that heparin binding may not be critical for signaling, although heparin may stabilize the FGF-FGFR complex formation. Also, heparin is known to bind to FGFs and non-specifically block integrin ligand interaction. Thus, even if heparin blocks FGF9-induced signaling, the effect may not be due to blocking FGF9 functions but due to blocking integrins. So, we will perform the suggested experiments using high concentration of heparin in future studies.
Minor points:
The part of the sentence "R108E suppressed DNA synthesis induced by WT FGF9 and suppressed DNA synthesis and activation of FRS2α and ERK1/2 induced by WT FGF9" appears to be a repetition (line 35-37). Response: Corrected.
No information about cyclic RGDfV in the Materials and methods section.
Response: the requested information was added.
It seems to me that in the results (3.2, line 213) in the description of the mutants there should be a reference to Fig. 3a instead of Fig. 1a.
Corrected.
The sentences " We found that R108E was defective in inducing cell migration (Figs. 4c and 4d). R108E was also defective in inducing invasion of DLD1 and Colon26 cells (Fig. 4e and 4f). These findings suggest R108E is defective in inducing migration and invasion in colon cancer cells" require rewording because it is a repetition (line 236-239).
Response: revised.
Figure 4b requires densitometric analysis.
Response: Fig. 4b was deleted. Fig. 4a and 4b have problems we cannot fix within the time limit. Also, these images have no new information that is critical for the paper.
How do the authors explain the lack of FGFR1 detection in the DLD1 cell line, which has been repeatedly known for the expression of this receptor? Why is only one band visible for FGFR2 and FGFR3, while these receptors usually present 2-3 forms? Can authors add a mass marker? (Fig. 4a, 4b).
Response: Fig. 4b was deleted. Fig. 4a and 4b have problems we cannot fix within the time limit. Also, these images have no new information that is critical for the paper.
In Fig. 5, how do the authors explain the different molecular weights of pFRS2 and FRS2? The shift may be due to phosphorylation, but then two bands should be visible in FRS2 detection, whereas only the one corresponding to the mass of the unphosphorylated protein is visible.
Response: In the original blotting data, p-FRS has two bands, and FRD has only a single band. Probably we selected only phosphorylated (upper band) for Fig. 5. We submitted the original images with two bands as a supplemental data (Fig. S1).
In my opinion, the intensity of the pErk band (Fig. 5a) does not correspond to the presented densitometric data (for the R108E mutein, last line on WB).
Response: The reviewer is right, but we cannot fix the issue now due to the limitation of time. We submitted the original images with two bands as a supplemental data (Fig. S1). It is possible that images may come from different blot (we have triplicate images). We have done more quantitative analysis from triplicate experiments (Fig. 5b).
Round 2
Reviewer 1 Report
Comments and Suggestions for Authors
I disagree with the justification of lack of time to repeat experiments but I appreciate the efforts of authors to answer to the comments.
Comments on the Quality of English Language
It has improved.
Author Response
I would like to response to the comment. We received the revision request just before holidays (around December 22, 2023) and I worked on the manuscript during holidays and sent back revised manuscript around new year. However, the lab was closed and was not able to work in the lab. We added new results on FGF9 binding to heparin and FGFR1. We also studied the effect of FGF9 mutant R108E on cell proliferation using two cancer cell lines in vitro. We have performed twice in anchorage-independent conditions but could not get publishable data. We will need to optimize experimental conditions such as incubation time, FGF9 concentrations, etc. We do not know in vitro half-life of FGF9. Also, we need to find optimum cell lines for the requested studies. We can probably transfect FGF9 gene into cancer cells. Anyway, we realized that it would take time to perform these experiments. My lab has just one research assistant due to funding shortage. We hope that the reviewer understands the situation. We will perform the requested experiments as a next step for the development of the FGF9 mutant as a potential therapeutic in future studies.